# Effect of Standoff Distance on Corrosion Resistance of Cold Sprayed Titanium Coatings

Mieczyslaw Scendo * 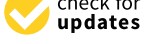 and Katarzyna Staszewska-Samson

Institute of Chemistry, Jan Kochanowski University in Kielce, Uniwersytecka 7, PL-25406 Kielce, Poland
* Correspondence: scendo@ujk.edu.pl

**Abstract:** A titanium protective layer was deposited onto Al7075 substrate (Al7075/Ti) by a cold spray method (CS) with different standoff distances (SoD) of the nozzle from the specimen surface. The aim of this research was to study the influence on the mechanical properties and corrosion resistance of the Ti coating on Al7075 substrate. The surface and microstructure of Al7075/Ti was observed by a scanning electron microscope (SEM). The corrosion test of the materials was carried out by using the electrochemical method. The SoD had a significant effect on the microstructure of the coatings and their adhesion onto Al7075 substrate. The highest level of microhardness (248 HV0.3) value was achieved for deposits obtained with the SoD of 70 mm. The corrosion tests showed that the mechanism of electrochemical corrosion of titanium coatings is a multi-stage process, and the main product of the corrosion process was $(TiO_2)_{ads}$. However, the polarization resistance ($Rp = 49\ k\Omega\ cm^2$) of the Al7075/Ti coatings was the highest, while the corrosion rate ($\upsilon_{corr} = 13.90\ mm\ y^{-1}$) was the lowest, for SoD of 70 mm.

**Keywords:** Al7075 substrate; standoff distance; cold spray; titanium coating; microstructure; corrosion rate

## 1. Introduction

Cold spraying (CS) is a coating technology based on aerodynamics and high-speed impact dynamics. In this process, spray particles (5–50 μm) are accelerated to a high velocity (300–1200 m/s) by a high-speed gas flow that is generated through a convergent–divergent de Laval type nozzle [1–3]. Pressurized gas ($N_2$ or He) is heated, typically by electric energy, to temperatures in the range of 300 °C to 800 °C and then directed to a nozzle to produce a supersonic inert gas stream [4]. Obtaining supersonic velocity by spray particles is a prerequisite for obtaining a coating that adheres well to the substrate, which ensures that it obtains the appropriate mechanical properties [5,6]. When the particles exit the nozzle and impinge on the target surface, they undergo significant plastic deformations resulting from collisions and bond to the substrate. Thus, there is a clear change in particle size [7]. A coating is formed through the intensive plastic deformation of particles impacting on a substrate at a temperature well below the melting point of the spray material. The main result is that CS can minimize effects of oxidation, melting, evaporation and other common problems suffered in thermal spraying [8]. Therefore, CS is used for the production and repair of metal coatings to increase mechanical properties and improve corrosion resistance of various metal components. However, it has been found that the particle size of the powder used to produce the coating has little effect on the deformation of the entire particle as it hits the substrate at high speed. It turned out that the particle flattening coefficient increases very markedly with the increase in the velocity of impact of the metal powder particles on the substrate surface. In contrast, the temperature at the localized contact surfaces increases significantly due to the possible adiabatic shearing process. The critical speed for particle deposition could be estimated with appropriate material properties [9,10]. The distance between the nozzle and the surface to which a new coating is applied has a

significant impact on the mechanical properties of metallic coatings. On the other hand, for cold spray deposition, increasing of the standoff distance (SoD) was generally thought to result in a lower particle speed due to the drag force exerted on the particle. Therefore, the protective coatings obtained did not have high mechanical parameters. On the other hand, too small a distance of the nozzle from the surface of the substrate results in lower quality coatings with low mechanical parameters. Therefore, a small SoD will reduce the particle velocity and deposition efficiency, which slows down the particle velocity before impact [11–19]. In order for the layer to be effectively formed on the surface of the substrate, the metal particles to be deposited must move at a supersonic speed clearly greater than the critical speed. It is worth adding that if the metal particle velocity is lower than the critical speed, the substrate may be significantly damaged as a result of its abrasion. Importantly, by increasing the powder feed rate in the nozzle, the velocity of the deposited metal particles is reduced due to gas–particle interactions as the particles move away from the nozzle outlet [20–22]. On the other hand, the production of metallic coatings is a multi-step process. Initially, a thin layer of material is deposited on the substrate. This stage is characterized by the direct interaction of the particles with the substrate and largely depends on the degree of preparation and the properties of the substrate. Subsequent layers are formed as a result of multiple overlapping coating material particles [23].

The aluminium–zinc–magnesium (i.e., Al7075) alloys have a greater response to heat treatment than the binary aluminium–zinc alloys, resulting in higher strengths. The additions of zinc and magnesium, however, decrease the corrosion resistance. Thus, the alloy Al7075 must be protected against corrosion, most often by means of metallic coatings resistant to corrosion. A possible approach to increase the corrosion resistance of this alloy is the deposition of a pure titanium coated layer. The titanium standard potential is $E_{Ti^{2+}/Ti} = -1.75$ V, so titanium is a non-noble metal. Titanium is a metal with high hardness and mechanical strength. Moreover, under natural conditions, the Ti surface undergoes a passivation, and therefore titanium is classified as a corrosion-resistant metal. However, the lower content of oxides is due to the cold spray being carried out at much lower temperature, and the reaction between the metal and the oxygen in the ambient spray can be greatly reduced or eliminated [24].

There is no reliable information in the literature so far concerning the influence of the distance between the nozzle and the substrate surface on the mechanical and anti-corrosion properties of metal coatings produced by the cold spray method.

The aim of this research was to check the impact influence of the different standoff distances of the nozzle from the specimen surface on the microhardness and corrosion resistance of the Ti coating onto Al7075 substrate. The titanium protective layer was produced by the cold spray method. The corrosion test was carried out by electrochemical method in an acidic chloride solution.

## 2. Experimental Details

### 2.1. Materials and Methods

The titanium coatings were deposited onto Al7075 substrate (Al7075/Ti) by the cold spray method (CS). Pure titanium powder (99.8 wt.% Ti) was used as feedstock. This powder was manufactured using the hydrite–dehydrite process and supplied by Kamb Import-Export (Warsaw, Poland). On the basis of the analysis of the powder's particle size distribution, it was found that was characterized by the d10 = 18.0 μm, d50 = 35.0 μm, and d90 = 60.0 μm [25], Figure 1.

The cold spray deposition was performed with an Impact Innovations 5/8 System (Impact-Innovations GmbH, Rattenkirchen, Germany). Nitrogen was applied as the process gas to the deposit titanium coatings. The maximum parameters used were a temperature of 800 °C and a pressure of nitrogen of 40 bar [25]. The cold spray equipment is shown in Figure 2.

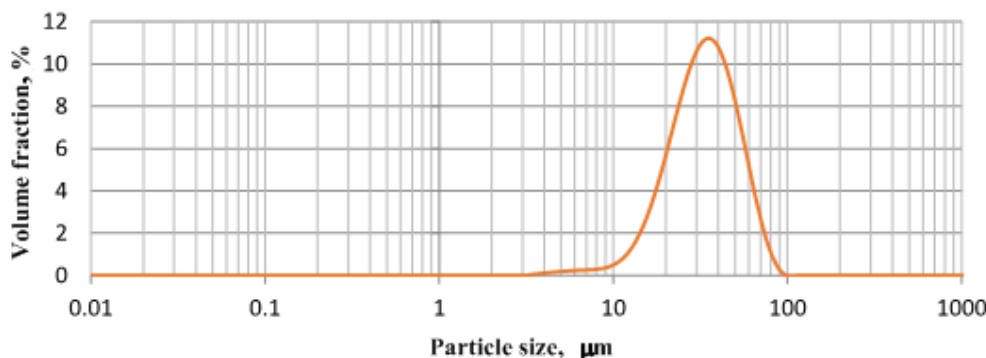

**Figure 1.** Particle size distribution of titanium powder [25].

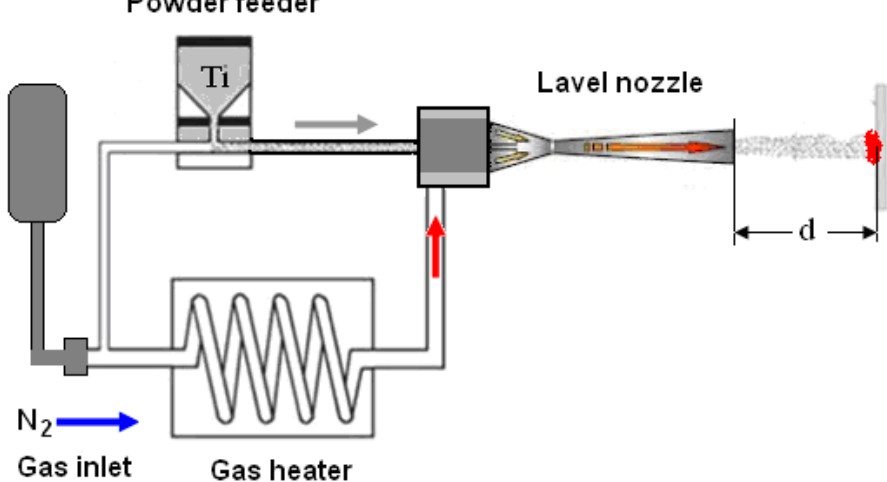

**Figure 2.** Typical cold spray equipment with a marked distance of the nozzle from the sample surface.

Before applying the Ti coating, the surface of the substrate, i.e., Al7075 was mechanically cleaned with 600, 1200, and 2000 grit sandpaper. Coatings were sprayed maintaining the distance (d) of the nozzle from the specimen surface in the range from 20 to 100 mm, which was systematically increased by 10 mm. The nozzle traverse speed was 400 mm/s. In order to obtain the planned thickness of the titanium coating, i.e., 1.5 mm–2 mm onto Al7075 surface, we made 2 or 3 runs with a cold spray gun [25]. The Al7075/Ti samples were cooled (in about 24 h) to room temperature under a nitrogen atmosphere. The basic parameters of the materials are listed in Table 1.

**Table 1.** Sample name, and distance of nozzle from sample surface.

| Sample | Distance d, mm |
|---|---|
| Al7075/Ti-20 | 20 |
| Al7075/Ti-70 | 70 |
| Al7075/Ti-100 | 100 |

Three of the Al7075/Ti samples were selected for the characterization of microstructures and corrosion test (Table 1).

### 2.2. Solutions

The following reagents were used to make the solutions: FLUKA analytical grade sodium chloride (NaCl) and POCH analytical grade hydrochloric acid (HCl). The concentration of $Cl^-$ ions was 1.2 M, and the pH was 1.5. The electrolyte was not deoxygenated.



*2.3. Electrodes*

The working electrode (W) was made of Al7075 alloy, which was covered with titanium coating (Al7075/Ti), which was produced by a cold spraying method. The geometric surface area of the W electrode was 1.0 cm$^2$. The saturated calomel electrode (SCE) was used as the reference. It was connected with the solution using a Luggin capillary. The counter electrode (9 cm$^2$) was made from platinum foil (99.9% Pt).

*2.4. Electrochemical Measurements*

The electrochemical experiments were carried out in a conventional three-electrode cell. All electrochemical measurements were made using a potentiostat/galvanostat PG-STAT 128N, (AutoLab, Amsterdam, The Netherlands) with NOVA 1.7 software from the same company.

The potentiodynamic polarization (LSV) curves were recorded. All measurements were carried out under a potential range from −900 mV to −100 mV vs. SCE, whereas the potential change rate was 1 mV s$^{-1}$, with holding time of 30 s at −900 mV.

The LSV curves were used to designate of the corrosion electrochemical parameters. However, the Stern–Geary equation was used for the calculation of the polarization resistance of the materials. The corrosion rate of materials were appointed using the following equation [26–29]:

$$v_{corr} = 3.268 \times \frac{j_{corr}\, M}{n\, \rho} \tag{1}$$

where $j_{corr}$ is the corrosion current density, $M$ is the molecular weight of the substrate, $n$ is the number of electrons exchanged, and $\rho$ is the density of the material.

The chronoamperometric curves (ChA) were obtained for the potential values which were selected for the characteristic points on the LSV curves.

*2.5. Surface Morphologies and Microstructure*

The surface morphologies and microstructure were observed by using a scanning electron microscope (SEM, JSM-5400, Joel, Tokyo, Japan). The accelerating voltage of SEM was 20 kV. Prior to the cross-sectional analysis, the coating samples were polished with increasingly fine, (3 μm, 1 μm, and 0.25 μm) diamond suspensions. The X-ray diffraction (XRD) was applied to characterize the phase composition of the coatings using a Bruker D8 Discover diffractometer (Bruker Ltd., Malvern, UK), with Co Kα radiation was of λ = 1.7889 Å.

*2.6. Microhardness*

The measurement of microhardness of the Al7075/Ti was made by the Vickers method (HV) using a Falcon 500 hardness tester from INNOVATEST (Maastricht, The Netherlands). An indenter was used in the form of a diamond pyramid, whose load varied from 0.02 N to 20 N.

All measurements were carried out at a temperature of 25 ± 0.5 °C which were maintained using an air thermostat.

**3. Results and Discussion**

*3.1. Surface Morphologies*

The surface morphologies of the Ti coatings onto Al7075 substrate are presented in Figure 3.

Titanium coatings adhere well onto Al7075 alloy. As shown in Figure 3, along with the change in SoD, a change in the surface structure of the Ti coating onto Al7075 substrate was observed. The most homogeneous and smooth surface of the Ti coating was obtained when the nozzle distance from the sample surface was 70 mm (Figure 3b).

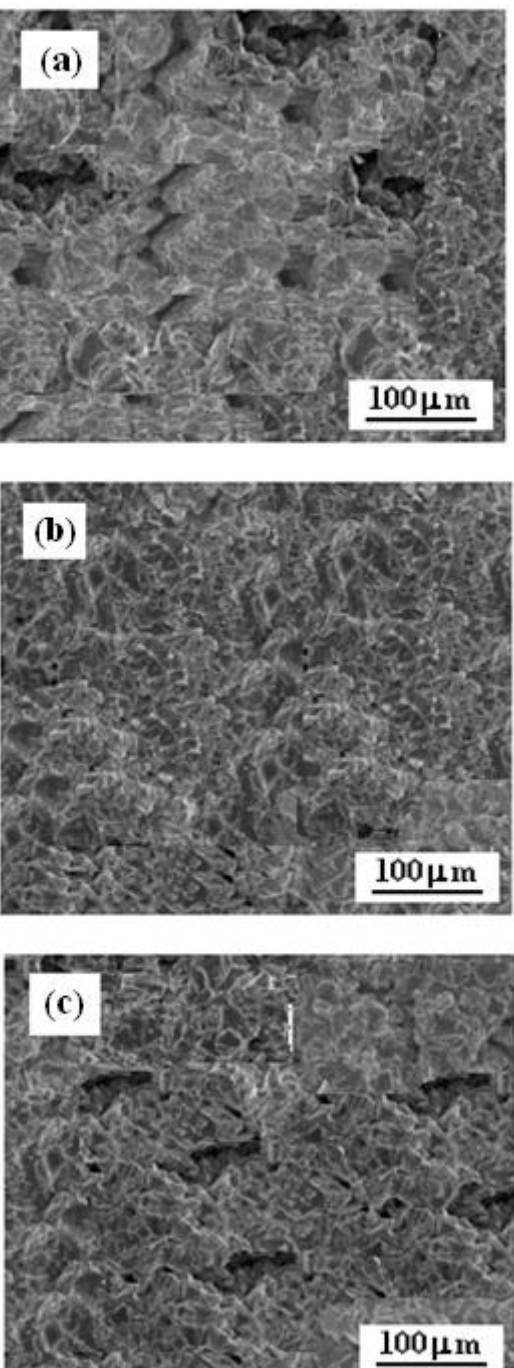

**Figure 3.** Surface morphologies of cold sprayed titanium coatings onto Al7075 substrate deposited at SoD of: (**a**) 20 mm, (**b**) 70 mm, and (**c**) 100 mm.

Figure 4 presents the X-ray diffraction patterns recorded for the Ti powder and cold sprayed coating deposited onto Al7075 substrate at SoD of 70 mm.

It can be seen that the SoD of 70 mm and high gas temperature of 800 °C do not cause the formation of oxides onto the Al7075/Ti surface. However, similar results were obtained for other samples (Table 1), but they are not cited in this work.

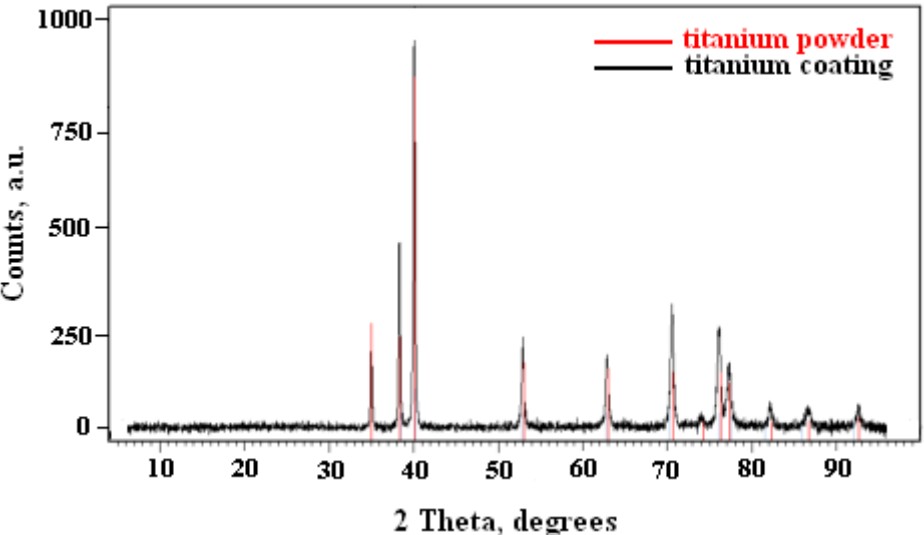

**Figure 4.** X-ray spectrum of cold sprayed titanium coating onto Al7075 substrate deposited at SoD of 70 mm.

### 3.2. Microhardness

In Table 2 shows how the SoD affects the microhardness of the Ti coatings onto Al7075 substrate. Similar observations were reported by [25].

**Table 2.** Microhardness of titanium coatings onto Al7075 substrate.

| Sample | Microhardness HV0.3 |
|---|---|
| Al7075/Ti-20 | $200 \pm 2$ |
| Al7075/Ti-70 | $248 \pm 1$ |
| Al7075/Ti-100 | $218 \pm 3$ |

The highest HV0.3 value was recorded for the titanium coating that was produced when the SoD was 70 mm (Table 2). The measurement results show a large deviation of the HV0.3 value, which is related to the different degree of deformation of the titanium particles on the Al7075 surface.

### 3.3. Corrosion Test

The corrosion tests of the Al7075/Ti coatings were carried out by the electrochemical method in 1.2 M Cl$^-$ solution (pH 1.5). The potentiodynamic polarization curves (LSV) of the titanium coatings onto the Al7075 alloy, depending on the nozzle distance from the sample surface, are shown in Figure 5.

The cathode branches (Figure 5) correspond to the simplified reduction reaction of hydrogen ions [26–28]:

$$Ti + 2\,H^+ \rightarrow Ti + H_2 - 2\,e^- \tag{2}$$

When the electrode potential is changed in the anode direction, the surface of the Ti electrode was oxidized:

$$Ti \rightarrow Ti^{2+} + 2\,e^- \tag{3}$$

Moreover, subsequently, an adsorbed oxide layer was formed on the surface of the electrode according to the reaction:

$$Ti^{2+} + 2\,H^+ + 3/2O_2 \rightarrow (TiO_2)_{ads} + H_2O + 8\,e^- \tag{4}$$

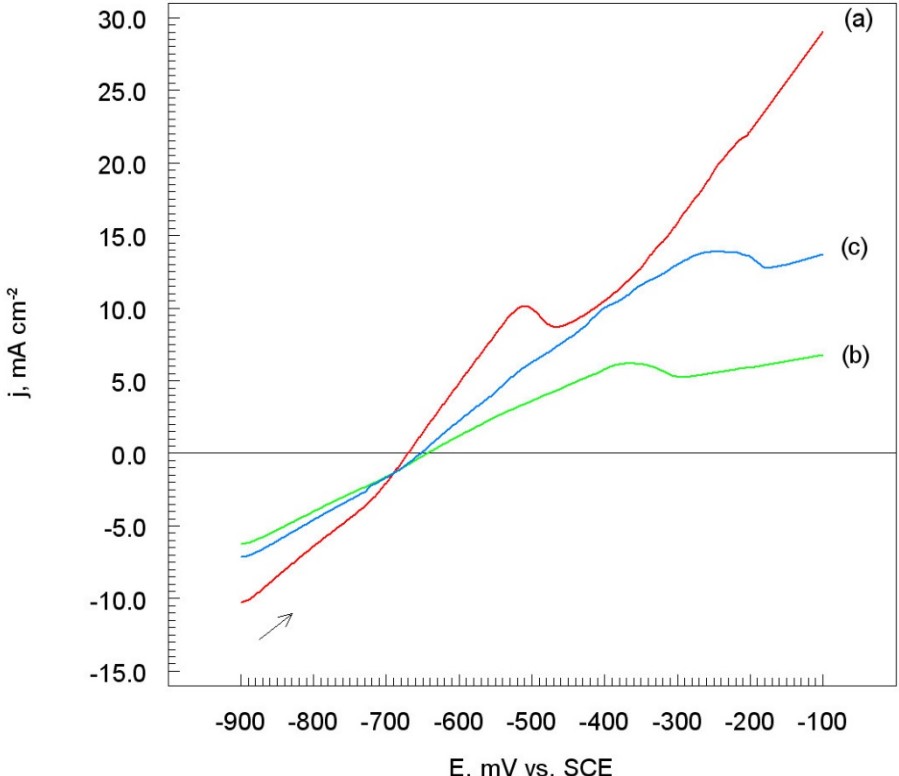

**Figure 5.** Potentiodynamic polarization curves of titanium coatings onto Al7075 substrate deposited at SoD of: (**a**) 20 mm, (**b**) 70 mm, and (**c**) 100 mm. Solutions contained 1.2 M Cl$^-$, pH 1.5, dE/dt 1 mV s$^{-1}$.

The Al7075/Ti surface was covered with a white layer of titanium (IV) oxide. Therefore, on the potentiodynamic polarization curves (Figure 5), peaks are observed for the electrode potentials of: −510 mV (curve (a)), −360 mV (curve (b)), and −230 mV (curve (c)) vs. SCE, respectively. Adsorbed titanium (IV) oxide (Equation (4)) clearly limits the effects of electrochemical corrosion on the titanium surface, especially for Al7075/Ti, which was produced when the nozzle distance from the sample surface was 70 mm (Figure 5, curve (b)). In this case, the protective oxide layer (reaction (4)) is tight and adheres well to the substrate, protecting the Al7075 substrate against contact with a corrosive chloride environment.

Under the conditions of the experiment in the acidic environment, the adsorbed oxide layer dissolves according to a simple chemical reaction:

$$(TiO_2)_{ads} + 4H^+ \rightarrow Ti^{4+} + 2H_2O \tag{5}$$

However, for more positive electrode potentials, the current density increases (Figure 5), hence a further dissolution of the titanium surfaces that have been deposited onto Al7075 substrate.

### 3.3.1. Corrosion Electrochemical Parameters

The LSV curves (Figure 5) were used to designate the corrosion electrochemical parameters of the Al7075/Ti coatings, i.e., corrosion potential ($E_{corr}$), corrosion current density ($j_{corr}$), and cathodic ($b_c$) and anodic ($b_a$) Tafel slopes (Figure 6).

The corrosion electrochemical parameters of the titanium coatings onto Al7075 substrate in aggressive chloride solution are listed in Table 3.

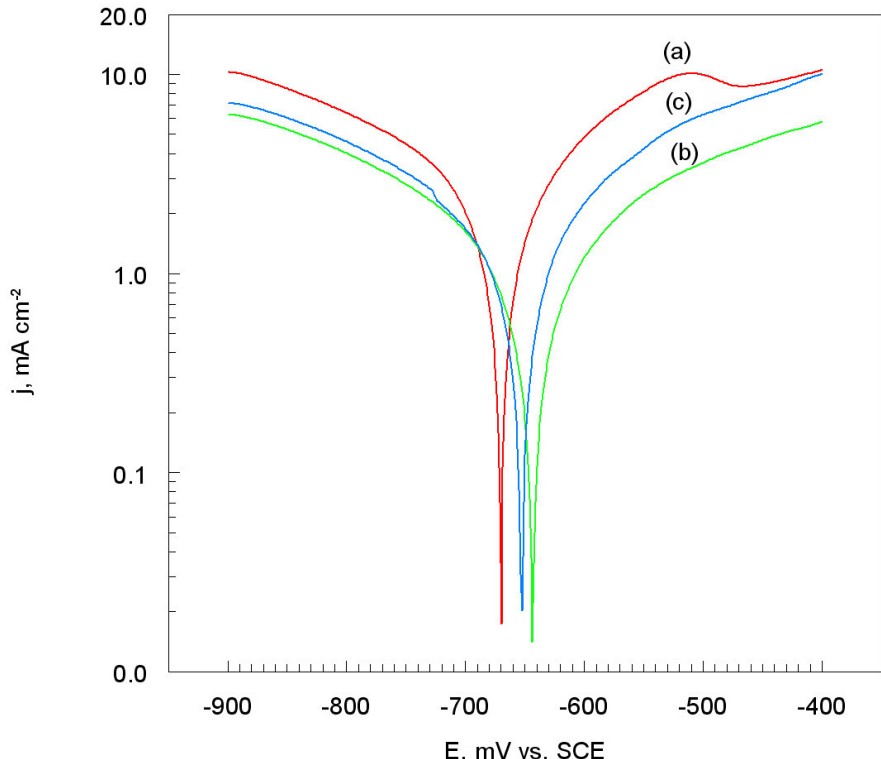

**Figure 6.** Potentiodynamic polarization curves on a semi-logarithmic (Tafel scale) of titanium coatings onto Al7075 substrate deposited at SoD of: (**a**) 20 mm, (**b**) 70 mm, and (**c**) 100 mm. Solutions contained 1.2 M Cl$^-$, pH 1.5, dE/dt 1 mV s$^{-1}$.

**Table 3.** Corrosion electrochemical parameters of titanium coatings onto Al7075 substrate.

| Sample | $E_{corr}$ mV vs. SCE | $j_{corr}$ mA cm$^{-2}$ | $-b_c$ | $b_a$ |
|---|---|---|---|---|
| | | | mV dec$^{-1}$ | |
| Al7075/Ti-20 | −670 | 2.80 | 350 | 250 |
| Al7075/Ti-70 | −643 | 1.60 | 340 | 380 |
| Al7075/Ti-100 | −652 | 1.90 | 380 | 300 |

It turned out that with the increase in the distance of the nozzle from the surface of the samples, the values of the corrosion potential shift towards positive values. Moreover, the most positive value of the corrosion potential was noted for the Al7075/Ti-70 coating (Table 3). Thus, if the distance between the nozzle and the surface of the sample was 70 mm, the Ti coating exhibits the best anti-corrosion properties. This is also evidenced by the lowest value of the corrosion current density (i.e., 1.60 mA cm$^{-2}$). The slopes of the Tafel's cathode fragments of polarization curves are slightly different between each other (Table 3). Therefore, the mechanism of cathodic hydrogen reduction (Equation (2)) is identical for the tested electrodes. However, the Tafel slopes of the anode segments of the polarization curves are different for each electrode material (Table 3). This suggests that the mechanism of electrochemical corrosion (Equations (3)–(5)) of the tested coatings depends on the method of Ti coating production onto Al7075 alloy (in this case, from the distance of the nozzle from the surface of the substrate).

3.3.2. Polarization Resistance and Corrosion Rate

The polarization resistance ($R_p$) of the titanium coatings were determined on the basis of the slope of the LSV curves (Figure 5). The corrosion rates ($v_{corr}$) of the materials were

calculated on the basis of Equation (1) [27–29]. The $R_p$ and $v_{corr}$ values of the titanium coatings cold sprayed onto Al7075 alloy are collected in Table 4.

**Table 4.** Polarization resistance and corrosion rate of titanium coatings onto Al7075 substrate.

| Sample | $R_p$ $k\Omega\ cm^2$ | $v_{corr}$ $mm\ y^{-1}$ |
|---|---|---|
| Al7075/Ti-20 | 23 | 24.30 |
| Al7075/Ti-70 | 49 | 13.90 |
| Al7075/Ti-100 | 38 | 16.50 |

The highest in the electrode's polarization resistance value was noted of the Ti coating, which was obtained when the nozzle distance from the surface of the substrate was 70 mm. Moreover, the electrochemical corrosion rate of this coating is also the smallest compared to Ti coatings that were produced by a different nozzle position (Table 4). It seems clear that metallographic structures of titanium coatings onto Al7075 alloy are completely different.

### 3.3.3. Chronoamperometric Measurements

Figure 7 shows the ChA curves in 1.2 M Cl$^-$ solution (pH 1.5) of the Ti coating onto Al7075 substrate deposited at SoD of 20 mm, 70 mm, and 100 mm. The electrode potential of the anode process was –150 mV vs. SCE, and was selected on basis of the LSV curves (Figure 5). It was found that, in the conditions of the experiment, the current density values depended on the distance of the nozzle from the surface of the substrate (Figure 7). For each nozzle distance from the Al7075 surface, the current density systematically decreases during electrolysis. Thus, the adsorbed oxide layer i.e., TiO$_2$ (reaction (4)) protects the Al7075 substrate against contact with an aggressive chloride solution.

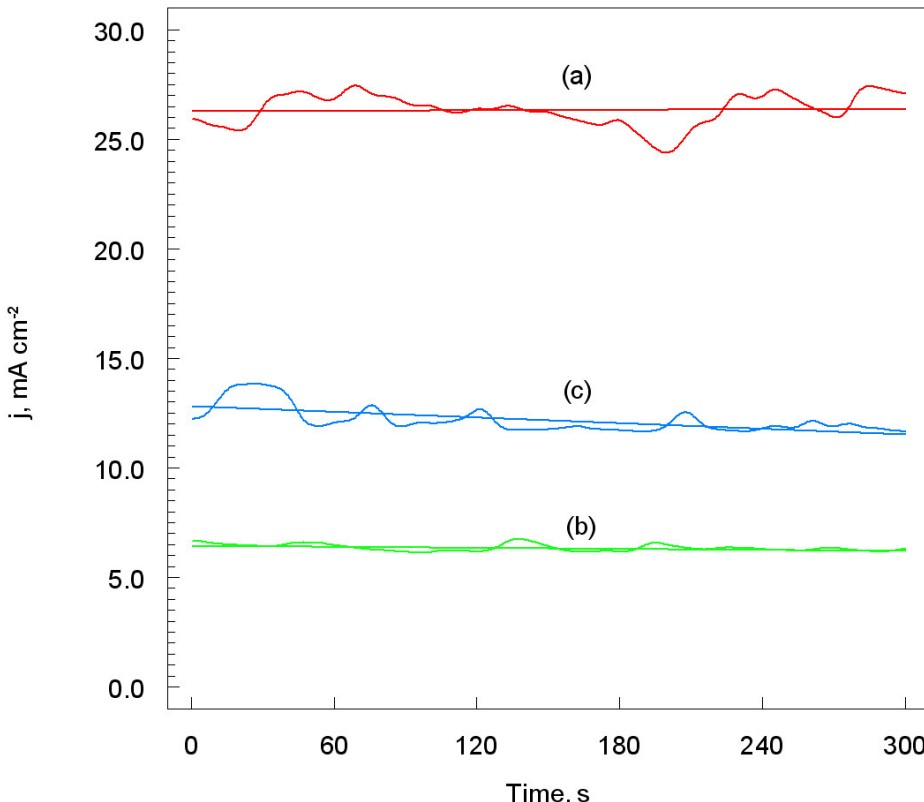

**Figure 7.** Chronoamperometric curves obtained for −150 mV of titanium coatings onto Al7075 substrate deposited at SoD of: (**a**) 20 mm, (**b**) 70 mm, and (**c**) 100 mm. Solutions contained 1.2 M Cl$^-$, pH 1.5. (Straight lines refer to the average current density values).

It seems that, under these conditions, the adsorbed oxide layer can be additionally sealed by adsorption of $Cl^-$ ions [29]:

$$MeO + Cl^- + H^+ \rightarrow (MeClOH^-)_{ads} \tag{6}$$

and

$$(MeClOH^-)_{ads} \rightarrow (MeClOH)_{ads} + m\ e^- \tag{7}$$

Unfortunately, the $(MeClOH)_{ads}$ layer in the acidic chloride solution was dissolved in accordance with a chemical reaction:

$$(MeClOH)_{ads} + H^+ \rightarrow Me^{n+} + Cl^- + H_2O \tag{8}$$

Thus, a further sharp increase in the current intensity was observed due to the oxidation of the electrode surface (Figure 5).

In addition, a clear fluctuation of the current density was observed (especially for the distance of 20 mm), which were associated with the oxidation of the surface of the test materials (Figure 7, curve (a)). Thus, in this case, the oxide surface is not homogeneous and does not effectively protect the substrate against corrosion.

Figure 8 shows a scheme of the formation of the protective titanium layer onto Al7075 substrate depending on the distance of the nozzle from the substrate surface.

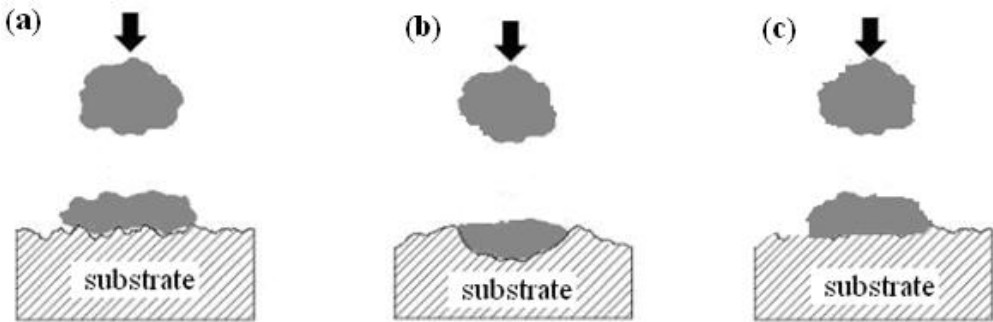

**Figure 8.** Scheme formation of titanium layer onto Al7075 substrate depending on nozzle distances from sample surface of: (**a**) 20 mm, (**b**) 70 mm, and (**c**) 100 mm.

During the production of the metal layer by the cold spray method, the particle impact speed must exceed the threshold value so that plastic deformation of the metal particles occurs, which causes them to adhere to the surface of the substrate. Moreover, nozzle geometry, powder characteristics and nozzle distance from the sample surface are of fundamental importance with respect to the microstructure, physical, and mechanical properties of the coatings.

If the distance of the nozzle from the substrate was 20 mm, the Ti particles were subjected to excessive deformation, which resulted in a leaky Ti coating onto Al7075 substrate (Figure 8a). It seems that the most compact and tight Ti coating was obtained when the distance of the nozzle from the substrate Al7075 was placed 70 mm. In this case, tightly adhering Ti coating was obtained (Figure 8b). However, during Ti coating production cold spray method when the distance of the nozzle from the substrate was increased to 100 mm, the Ti particles did not deform sufficiently and a leaky porous coating was obtained (Figure 8c). The aggressive chloride electrolyte easily penetrates deep into the structure of the Ti coating, causing corrosion of the substrate.

On the other hand, an important aspect of the powder morphology is that the particle shape of the coating material has a significant influence on the mechanical properties of the newly formed coating. It turns out that irregularly shaped metal particles hitting the substrate cause high internal stresses in the new coating. Ajdelsztajn et al. [30] showed that when the shell metal particles hit the substrate, a localized shear deformation occurs at the particle boundaries. In this way, the shape of the metal particles is changed to

be more regular, which favors close contact between the particles and helps to form a metallurgical particle/particle bond. In addition, the irregular morphology of the metal particles will increase the concentration of internal stresses in the coating due to the fact that the load cannot be evenly distributed. Thus, the regular morphology of the coating material particles promotes an even distribution of internal stresses, and thus significantly improves the mechanical properties of the new coatings [28]. Moreover, the concentration of stresses on the surface of the deposited metal particles may facilitate the occurrence of a localized shear deformation that can cause microcracks on the surface of the newly produced metallic coating.

### 3.4. Microstructure Titanium Coatings

The SEM microstructure of the cross-section of the titanium coatings onto Al7075 substrate (for the of SoD of 20 mm, 70 mm, and 100 mm) before and after exposure in 1.2 M $Cl^-$ solution (pH 1.5) are shown in Figure 9. The exposure time was 4 h.

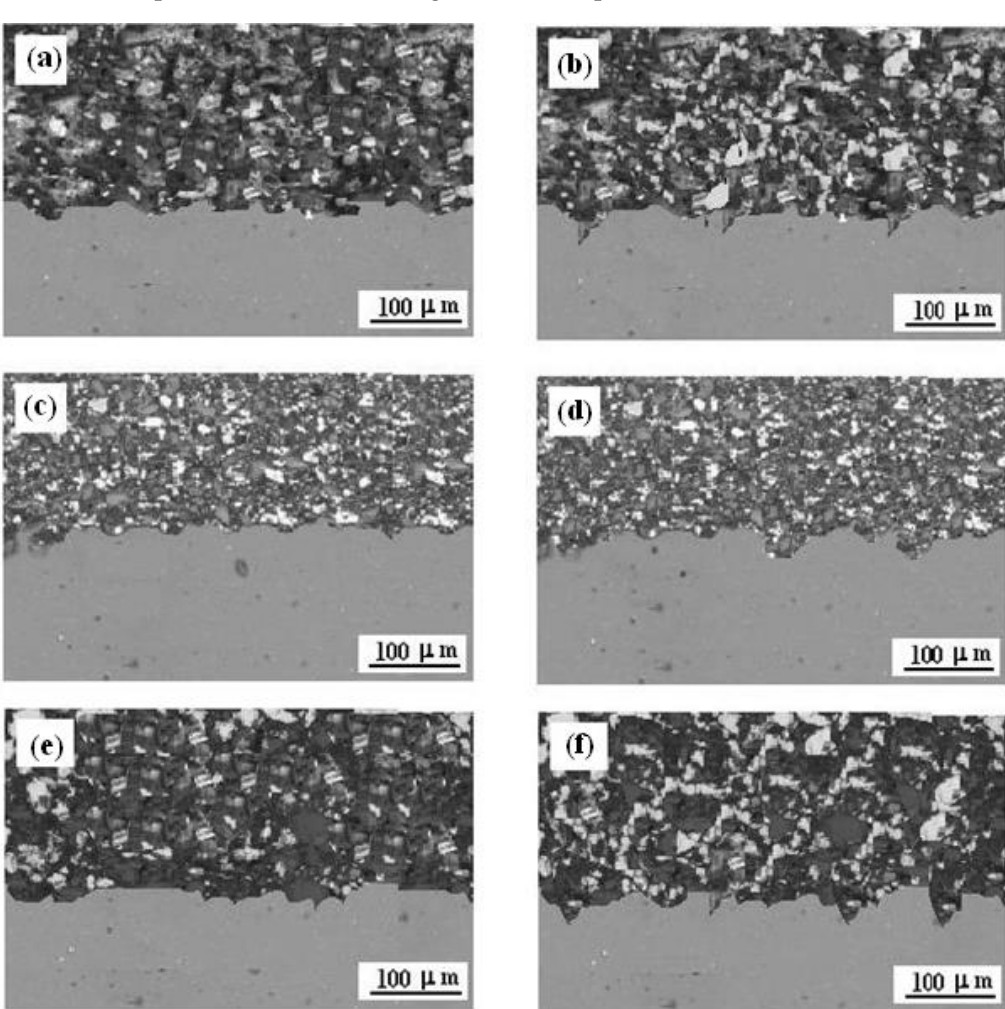

**Figure 9.** SEM of cross-section of titanium deposited onto Al7075 substrate at the SoD of: (**a**) 20 mm, (**c**) 70 mm, and (**e**) 100 mm, and after exposure at 1.2 M $Cl^-$ (pH 1.5) at SoD of: (**b**) 20 mm, (**d**) 70 mm, and (**f**) 100 mm. Exposure time was 4 h.

Significant changes in the microstructure of the Al7075/Ti coatings, together with increasing of the SoD in the range of 20 mm to 100 mm, are visible. Spraying at short distances (20 mm) produces porous of Ti coatings which adhere well to the Al7075 alloy substrate (Figure 9a). The reason for such increased porosity of the coatings in these cases may be the lower speed of larger particles of the titanium powder. After leaving the nozzle, the velocity of the Ti powder grains increases depending on their size, reaching the highest

value at a distance of 70 mm or 100 mm [25]. A close SoD is sufficient for large particles to form good adhesion with the substrate, with no pores visible at the interface. Extending the distance of the nozzle from the substrate leads to a noticeable reduction in the porosity, which reaches the minimum value at 70 mm, which is clearly visible in Figure 9c. However, this coating is characterized by a small number of very fine pores that appear evenly over the entire cross-sectional area.

Further increasing the distance of the nozzle from the substrate material, i.e., 100 mm, resulted in a significant increase in the porosity of the Ti coating onto Al7075 substrate (Figure 9e). The large distance of the nozzle from the substrate resulted in a decrease in the velocity of all the grains of the Ti powder, which resulted in an increase in the porosity of the Al7075/Ti coating. The most significant decrease in velocity concerned fine Ti powder grains, which deformed slightly, contributing to a significant increase in the porosity of the coating [31]. The kinetic energy of the largest Ti grains was so high that after impact with the previously applied layer of titanium, the coating deformation process took place. Such changes in the porosity of the coatings sprayed with cold gas result from the course of the coating forming process, where the last layer is not subjected to penning as intensively as the layers previously deposited [32,33].

The cross-section of the titanium coatings onto Al7075 substrate (for the SoD of 20 mm, 70 mm, and 100 mm) after exposure (4 h) in chloride solution are shown in Figure 9b–f. As a result of the corrosive action of the environment, the greatest destruction of the Ti layer was observed in the case of Al7075/Ti-20 and Al7075/Ti-100 samples (Table 1). It is clearly visible that the inhomogeneous, porous structure of the Ti coatings on the aluminum alloy substrate (Figure 9b,f) were degraded as a result of the reaction of the coating with the corrosive environment. The numerous pores on the titanium surface cause the electrolyte to penetrate deep into the Ti layer, which causes intense electrochemical corrosion of the aluminum substrate (reactions (6)–(8)). However, in the case of the SoD of 70 mm, a fairly smooth, compact structure of the Al7075/Ti coating was obtained. Therefore, the titanium surface was slightly damaged (Figure 9d), still protecting the Al7075 substrate from the corrosive action of the acid chloride solution.

## 4. Conclusions

The work carried out presents experimental studies of the titanium deposition onto Al7075 substrate by cold spraying at different distances between the nozzle and the substrate. The microstructure and anti-corrosion properties of the Ti onto Al7075 alloy in an acid chloride solution were investigated. The conducted research allowed for the formulation of the following conclusions:

1. The different standoff distance of the nozzle from the specimen surface (i.e., 20 mm, 70 mm, and 100 mm) in the cold spray process has a significant influence on the properties of the Al7075/Ti coating.
2. Titanium coatings adhere well onto Al7075 alloy, and the most homogeneous and smooth surface of the Ti coating was obtained when the nozzle distance from the sample surface was 70 mm.
3. The microhardness (HV0.3) of the deposit depend significantly on the nozzle distance. The highest level of HV0.3 value was achieved for deposits obtained with the SoD of 70 mm.
4. There were no phase changes in the phase composition of the titanium deposits due to the increased of SoD.
5. Corrosion test (electrochemical method) of the titanium coatings onto Al7075 substrate were carried out in acidic chloride solutions.
6. The mechanism of electrochemical corrosion of titanium coatings is a multi-stage process, and the main product of the corrosion process was $(TiO_2)_{ads}$. The oxide layer did not protect the materials against the penetration of the aggressive solution.
7. The polarization resistance ($R_P$) of the Ti coatings was the highest, while the corrosion rate ($v_{corr}$) was the lowest, for the SoD of 70 mm. Thus, in this case, the exchange

of mass and electrical charge between the electrode and the electrolyte solution is significantly impeded.

8. The titanium surface on the Al7075 substrate was slightly damaged when exposed to an acid chloride solution, while still protecting the aluminum substrate from the corrosive effects of the environment.

**Author Contributions:** Conceptualization, M.S.; formal analysis, M.S. and K.S.-S.; investigation, K.S.-S.; methodology, M.S.; writing-original draft, M.S. and K.S.-S.; writing-review and editing, M.S. All authors have read and agreed to the published version of the manuscript.

**Funding:** The work reported herein has been undertaken as part of the project funded by the Jan Kochanowski University in Kielce, ID: SUPB. RN. 21.186.

**Institutional Review Board Statement:** Not applicable.

**Informed Consent Statement:** Not applicable.

**Data Availability Statement:** Data sharing is not applicable.

**Conflicts of Interest:** The authors declare no conflict of interest.

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
