# Peer review of "Effect of Standoff Distance on Corrosion Resistance of Cold Sprayed Titanium Coatings"

_coatings, doi:10.3390/coatings12121853_

Round 1
Reviewer 1 Report
In this paper, the cold spraying method (CS) was used to deposit a titanium protective layer on the Al7075 substrate. The effects of the distance between the nozzle and the sample surface on the microstructure, adhesion, hardness and corrosion resistance of the coating were studied. There are the following problems:
1. Part3.1 "Titanium coatings adhere well onto Al7075 alloy." The author only provided the surface morphology diagram here, but the adhesion of the coating to the substrate should be assessed by applying the section morphology diagram.
2. The properties of materials are determined by the structure of materials. The author only piles up the data of properties characterization in the paper, and lacks mechanical explanation, such as whether the cause of the change of coating hardness is related to its internal stress. The internal stress should be characterized, whether it is related to the grain size, the writing idea of the article is confused, should be from how different spray distance affects the internal microscale change of coating, thus affecting its performance is different, so the author needs to make writing adjustment.
3. There are some obvious negligent errors in the paper, such as picture serial number.
Author Response
Reviewer 1
Thank you very much for the thorough and substantive review of our article. Below I have answered the questions and suggestions in the review of our work.
There are the following problems:
1.Part3.1 "Titanium coatings adhere well onto Al7075 alloy." The author only provided the surface morphology diagram here, but the adhesion of the coating to the substrate should be assessed by applying the section morphology diagram.
The scanning electron microscopy (SEM) microstructure of the cross-section of titanium coatings on the Al7075 substrate (for a distance of 20 mm, 70 mm, 100 mm) is shown in Figures 9a, 9c, and 9e (as new). Moreover, the problem of adhesion of Ti coatings to the Al7075 substrate was thoroughly discussed.
- The properties of materials are determined by the structure of materials. The author only piles up the data of properties characterization in the paper, and lacks mechanical explanation, such as whether the cause of the change of coating hardness is related to its internal stress. The internal stress should be characterized, whether it is related to the grain size, the writing idea of the article is confused, should be from how different spray distance affects the internal microscale change of coating, thus affecting its performance is different, so the author needs to make writing adjustment.
I agree with the reviewer's opinion that we have not discussed some important issues in this article. Some of them, e.g. the problem of internal stresses, are currently being investigated by us and will be discussed in detail in the next article.
Reviewer 2 Report
The authors researched the standoff distance on corrosion resistance of CS Titanium coatings. However, I can not see any novel works.
Firstly, authors declared that the CS is a novel technique. However, CS is quite a mature and widely used technique.
Besides, the authors want to research the effect of standoff distance on the corrosion resistance of coatings. They have not design a good experiment about it.
The structure and logical of this manuscript is quite poor. Thus, I must give it reject.
Author Response
Reviewer 2
Thank you very much for the thorough and substantive review of our article.
The authors researched the standoff distance on corrosion resistance of CS Titanium coatings. However, I can not see any novel works.
Firstly, authors declared that the CS is a novel technique. However, CS is quite a mature and widely used technique.
Besides, the authors want to research the effect of standoff distance on the corrosion resistance of coatings. They have not design a good experiment about it.
The structure and logical of this manuscript is quite poor. Thus, I must give it reject.
We're sorry you didn't like our article.
Reviewer 3 Report
The following, some points should be revised before I can recommend the paper for publication.
1. The abstract should be revised specifying the context of the research, the gap of knowledge, the aim of the work, the materials and the methods adopted and the main results obtained.
2. The novelty of the work is not clear from Introduction. Which is your main scientific contribution? Can you revise this, please?
3. The authors say within Introduction: “In turn, too small distance between the nozzle and the surface of the substrate, which is contained from 2.5 cm to 6 cm, makes the surface of the coating not homogeneous and uneven, and are characterized by low mechanical parameters”. Can you explain shortly the reasons of such phenomena? In other words, why does the surface of the coating appear not homogeneous and uneven?
4. In my opinion, the authors do not give significant emphasis to cold spray technology within Introduction, they do not describe in details the process, the advantages to be “Cold” and the mechanisms ruling the bonding that are still open questions in literature. The reviewer suggests to improve the manuscript with the missing parts and to add some references on CS within the introduction section. I can suggest the following references that describe the process, the major applications, the issues, the advantages and the need of developing numerical models to study CS in more details.
a. Experimental study and numerical investigation of the phenomena occurring during long duration cold spray deposition, Int. Rev. Model. Simulations. 11 (2018) 84–92. https://doi.org/10.15866/iremos.v11i2.13619.
b. Experimental evaluation of metallic coating on polymer by cold spray, in: Procedia Manuf., Elsevier B.V., 2020: pp. 761–765. https://doi.org/10.1016/j.promfg.2020.04.232.
c. Some aspects on 3D numerical modeling of high velocity impact of particles in cold spraying by explicit finite element analysis, Appl. Surf. Sci. 255, 7878-7892 (2009).
d. Particle Bonding Mechanism in Cold Gas Dynamic Spray: A Three-Dimensional Approach, J. Therm. Spray Tech. 26, 1859-1873 (2017).
e. Computational Analysis of the Interfacial Bonding Between Feed Powder Particles and the Substrate in the Cold-Gas Dynamic-Spray Process, Appl. Surf. Sci. 219(3-4), 211-227 (2003).
f. On the influence of different superficial laser texturing on the deposition of powders through cold spray process, Trans. Inst. Met. Finish. 96 (2018) 34–40. https://doi.org/10.1080/00202967.2018.1403096.
5. Figure 2 is not necessary and it does not provide a significant contribution at the paper. The scheme of High Pressure Cold Spray is known in literature.
6. What do you mean with “deposition step size” in cold spray?
7. Can you describe the strategy of manufacturing of the coating? How many passes of the cold spray gun did you perform? Which is the pattern used?
8. Is it possible to provide more information about CS parameters selection? Was any statistical method used to optimize the coating?
9. You say that: “As depicted in Figure 3 , an increase in the SoD resulted in a decrease in porosity and an improvement in the mechanical properties of the material”. Did you measure the porosity? Can you quantify the porosity?
10. It should be necessary showing SEM images of the particle impact at different SoD distances to prove the scheme reported in Fig. 8.
11. Careful reading of the text should be done in order to suppress typo errors; can you check them, please?
Author Response
Reviewer 3
Thank you very much for the thorough and substantive review of our article. Below I have answered the questions and suggestions in the review of our work.
The following, some points should be revised before I can recommend the paper for publication:
- 1. The abstract should be revised specifying the context of the research, the gap of knowledge, the aim of the work, the materials and the methods adopted and the main results obtained.
Following the reviewer's recommendation, I edited the Abstract again.
- The novelty of the work is not clear from Introduction. Which is your main scientific contribution? Can you revise this, please?
Indeed, in the text of the Introduction, there was no information regarding the influence of the distance between the nozzle and the substrate on some properties of the coatings produced by the cold spray method. So in the Introduction section I added the following information:
There is no reliable information in the literature so far concerning the influence of the distance between the nozzle and the substrate surface on the mechanical and anti-corrosion properties of metal coatings produced by the cold spray method.
- The authors say within Introduction: “In turn, too small distance between the nozzle and the surface of the substrate, which is contained from 2.5 cm to 6 cm, makes the surface of the coating not homogeneous and uneven, and are characterized by low mechanical parameters”. Can you explain shortly the reasons of such phenomena? In other words, why does the surface of the coating appear not homogeneous and uneven?
Thank you for pointing out the wrongly worded sentence. The new sentence reads as follows:
On the other hand, too small distance of the nozzle from the surface of the substrate results in lower quality coatings with low mechanical parameters.
- In my opinion, the authors do not give significant emphasis to cold spray technology within Introduction, they do not describe in details the process, the advantages to be “Cold” and the mechanisms ruling the bonding that are still open questions in literature. The reviewer suggests to improve the manuscript with the missing parts and to add some references on CS within the introduction section. I can suggest the following references that describe the process, the major applications, the issues, the advantages and the need of developing numerical models to study CS in more details.
Thank you for your valuable comment on important information related to the cold spray method. I used the recommended articles and supplemented the Introduction text with additional information on the described method of producing metal coatings.
- Figure 2 is not necessary and it does not provide a significant contribution at the paper. The scheme of High Pressure Cold Spray is known in literature.
I agree with the reviewer that Figure 2 depicts Typical cold spray equipment that is easily found in the literature. However, I would like to leave this Fig. 2. as it clearly shows the distance between the nozzle and the sample surface.
- What do you mean with “deposition step size” in cold spray?
Indeed, this part of the sentence was incorrectly edited. Therefore, I deleted this part of the sentence.
- Can you describe the strategy of manufacturing of the coating? How many passes of the cold spray gun did you perform? Which is the pattern used?
The cold spray deposition was performed with an Impact Innovations 5/8 System (Impact-Innovations GmbH, Rattenkirchen, Germany), cooperating with a Fanuc M-20iA robot (Fanuc Robotics Ltd., Oshino, Japan). In order to obtain the planned thickness of the titanium coating (1.5 mm - 2 mm) on the Al7075 surface, we made 2 or 3 runs with a cold spray gun.
- Is it possible to provide more information about CS parameters selection? Was any statistical method used to optimize the coating?
The basic parameters for the production of Ti/Al7075 coatings using the CS method are provided in section 2.1. Materials and Methods. Unfortunately, we did not use any statistical method to optimize the coating.
- You say that: “As depicted in Figure 3 , an increase in the SoD resulted in a decrease in porosity and an improvement in the mechanical properties of the material”. Did you measure the porosity? Can you quantify the porosity?
The reviewer was right. The above sentence is poorly drafted. Therefore, I have replaced them with a new sentence that reads as follows:
As shown in Figure 3, along with the change in SoD, a change in the surface structure of the Ti coating onto Al7075 substrate was observed.
- It should be necessary showing SEM images of the particle impact at different SoD distances to prove the scheme reported in Fig. 8.
In Figure 8, I only wanted to present a diagram of the formation of a protective titanium layer on the Al7075 substrate depending on the distance of the nozzle from the substrate surface. However, in Figure 9, I have presented and discussed the corresponding SEM pictures changes in the microstructure of the Al7075/Ti Coatings together with increasing of the standoff distance of the nozzle in the range of 20 to 100 mm.
Round 2
Reviewer 1 Report
The article was written in confusion, and the author avoided answering the questions raised, so it is recommended to reject the draft.
Author Response
Response to Reviewer 1 Comments
Point 1: The article was written in confusion, and the Author avoided answering the questions raised, so it is recommended to reject the draft.
Response 1:
In my opinion, I answered the questions asked in the review in accordance with the results of the research. I agree with the reviewer that some problems with changing the mechanical properties of Al7075/Ti coatings due to the use of different nozzle distances from the substrate are not fully explained. Among other things, the influence of internal stresses on the mechanical properties of the coatings should be discussed. Unfortunately, we do not have the appropriate equipment and at the moment we do not have the necessary test results. We will investigate this problem together with colleagues from the Kielce University of Technology and discuss it in the next publication.

Reviewer 2 Report
The authors have improved the quality of this manuscript. It can be accepted after minor revision.
1. The SEM figures can be merged together in one line for a better reading.
2. The TiO2 maybe reduction by H+ in the reaction (5).
Author Response
Response to Reviewer 2 Comments
Point 1: The SEM figures can be merged together in one line for a better reading.
Response 1:
The SEM photos (Figure 9) are compiled in the same order as the distance (SoD) between the nozzle and the substrate surface changed, i.e. 20 mm, 70 mm, and 100 mm. (SEM of cross-section of titanium deposited onto Al7075 substrate at the SoD of: a) 20 mm, b) 70 mm, and c) 100 mm, and after exposure at 1.2 M Cl- (pH 1.5) at SoD of: b ) 20 mm, d) 70 mm, and f) 100 mm). In my opinion, a different positioning of the SEM photos is possible, but it can make it difficult for the reader to follow the Al7075/Ti structure changes as the distance of the nozzle from the substrate.
Point 2: The TiO2 maybe reduction by H+ in the reaction (5).
Response 2:
In my opinion, under the conditions of the experiment, the titanium oxide (TiO4) adsorbed on the electrode surface in an acidic environment (pH 1.5) undergoes a simple chemical reaction according to Equation (5). Therefore, we did not detect of the TiO4 on the electrode surface (Figure 4).

Reviewer 3 Report
Can the authors describe better the pattern used for deposition, please?
The measurement of the porosity is necessary to analyse the effectiveness of the produced coatings.
Author Response
Response to Reviewer 3 Comments
Point 1: Can the Authors describe better the pattern used for deposition, please?
Response 1:
All information about the production of Al7075/Ti coatings that I received from the manufacturer was included in point 2.1. Materials and Methods. Unfortunately, I was unable to get more details on the other parameters that were used in the production of Al7075 / Ti coatings.
Point 2: The measurement of the porosity is necessary to analyse the effectiveness of the produced coatings.
Response 2:
I agree with the reviewer that measuring the surface porosity of Al7075/Ti samples is very necessary. In this way, the anti-corrosion properties of titanium coatings can be thoroughly discussed. Unfortunately, we do not have the appropriate equipment to quantify titanium coatings on the Al7075 substrate. Therefore, to discuss the results, we used the qualitative concept of the surface porosity of the coatings, using SEM of the Al7075/Ti cross-section (Figure 9).
